# MFCL: A Multi-modal Function Calling Evaluation for Large Language Models

## Abstract

Large language models are evolving into multi-modal agents that call tools directly from raw speech or images. Yet we still lack a principled metric for how well they convert perception into accurate function calls. We introduce **MFCL**, the first large-scale benchmark for *Multi-modal Function Calling*, comprising **8.2K** expert-verified tasks across three suites—**True Audio**, **Text Audio**, and **Vision**. Each example pairs a multi-modal user query with a ground-truth tool-call trace. To examine different capabilities of the LLM's perception-to-action pipeline, we introduce controlled perturbations: for audio, accents, contractions, simplified forms, casual pronouns, slang, disfluencies (fillers, hesitations, repetitions), and background noise; for images, crops and resizes, occlusions, grayscale and other color shifts, and related transformations. Image crops and resizes, occlusions, black-and-white and other color filters, etc for images. Our automatic grader computes exact-match scores for both function names and their arguments, removing dependence on brittle LLM judges and isolating errors in perception, reasoning, and formatting. We evaluate leading models and present a taxonomy of failure models: named-entity ASR errors, conversational drift, and tool avoidance. By releasing MFCL's dataset, taxonomy, and diagnostics, we hope to accelerate research on multi-modal agents that can effectively invoke tools.

## 1 Introduction

Large Language Models (LLMs) have rapidly transitioned from pure text interfaces to *tool-augmented* agents capable of calling external functions such as database look–ups, API endpoints, or robotic controllers. Recent releases from leading labs and open-source communities—such as GPT 4o, Gemini 2.5 Pro, and Llama 4—have extended this capability beyond text: a single model can now listen, watch, and speak, invoking the same JSON-style function calls from raw audio or images. Despite the commercial excitement, we lack a systematic evaluation of how well existing multi-modal models actually perform *end-to-end function calling*.

Current benchmarks focus on either (i) text-only tool use—e.g., BFCL(Patil et al., 2025), T-Eval(Chen et al., 2024), and $\tau$-BENCH(Yao et al., 2024)—or (ii) general multi-modal understanding such as MMMU(Yue et al., 2024). None measure the specific failure modes that arise when acoustic noise corrupts an automatic speech recognition (ASR) stage, when visual occlusion hides a key argument, or when the model produces fluent, conversational text instead of using tools. Without such diagnostics, it is impossible to decide whether an error stems from perception, reasoning, or formatting—and therefore impossible to improve the system in a targeted manner.

We introduce **MFCL** (Multi-modal Function Calling Evaluation), the first benchmark to fill this gap and evaluate end-to-end *multimodal function calling*. MFCL is a single framework containing *8.2K* diverse tasks, spanning over three eval suites: MFCL True Audio, MFCL Text Audio, and MFCL Vision. Each task specifies (1) a user request presented in transcribed text, speech, or an image, (2) a ground-truth JSON function call, and (3) a rich set of distractors such as background noise, accent variety, or image crops that stress different parts of the tool-calling pipeline. Our annotation protocol yields **exact-match** references at the granularity of both function name and argument values, enabling automatic grading without relying on fragile LLM judges.

Our analysis of the audio suites uncovers three recurring failure modes: (i) **Incorrect Named-Entity Recognition**, where mis-hearing a single proper noun invalidates the call; (ii) **Clarification Failures**, where models ask unsupported follow-up questions; and (iii) **Conversational Drift**, where

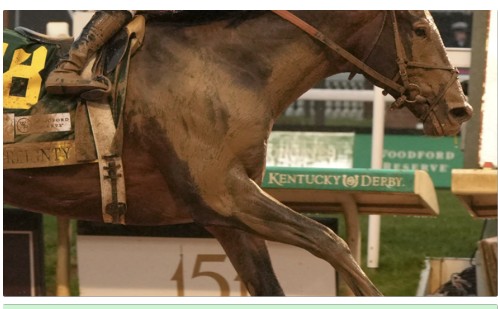

**Query:** I grew up going to horse races with my dad and recently attended this one. We always make a fun bet on a horse between ourselves. I snapped this picture of the horse I chose, but I can't remember where it finished. Can you tell me what place it came in?

**Reasoning Trace:**
- Identify text → "Kentucky Derby", "151".
- Extrapolate that this is the 151st Kentucky Derby.
- Identify text on the horse's sash → "REIGNTY"
- Search "151st Kentucky Derby Rankings" and find a site with the list of rankings with each horse's name.
- Traverse rankings site for string match "REIGNTY"
- Find that this horse is named Sove**reignty** and it placed 1st.

**Answer:** 1st place

Figure 1: Example entry from the MFCL Vision suite. The model must interpret both textual and visual cues (e.g., the event banner, race number, and horse sash) to ground a `web-search` tool call that retrieves the external information necessary to answer the question. MFCL Vision tasks are intentionally designed to go beyond simple visual reasoning, stressing perception–tool-use capabilities under real-world noise, occlusion, and distractors. Unlike prior multi-modal datasets, MFCL Vision evaluates whether models can robustly translate visual understanding into precise tool calls that lead to accurate responses.

RLHF-tuned models output polite chat instead of JSON tool calls—echoing findings from JSON-former (Liu et al., 2023) and hallucination audits (Gudibande et al., 2023; Sun et al., 2023).

The vision suite shows a different pattern: (i) **Avoiding Tool Use**, in which models abstain or ask questions instead of issuing a search; (ii) **Poor Keyword Selection**, where generated queries are vague or irrelevant; and (iii) **Visual Reasoning Errors**, ranging from misreading text to overly generic attribute descriptions. To (a) disentangle perception from reasoning, and (b) study the effectiveness of different image-training paradigms, we run controlled ablations on the image, including grayscale conversions, canny-edge filters, color jittering, and partial occlusion—and track. Through our study, we formulate and present the first error taxonomy to guide future research in this field of multi-modal tool-calling.

This paper makes the following primary contributions:

1. We propose **MFCL**, the first benchmark to systematically evaluate *multi-modal* function calling (aka. tool use) in LLMs under real-world acoustic and visual perturbations.

2. We curate and release *8.2K* tasks spanning text, audio, and vision with ground-truth references and automated grading.

3. We conduct a large-scale study of models and identify and analyze dominant failure modes, providing actionable insights to accelerate progress toward reliable LLM agents.

With our comprehensive harness, we hope MFCL will become a standard framework for multi-modal tool-call evaluation.

## 2  RELATED WORK

In this section, we examine related work on tool-calling evaluation of LLMs.

**Tool-Calling Benchmarks:**   Given the growing recognition of tool-calling (also called function-call), various works have proposed evaluation techniques and benchmarks. Early benchmarks such as TOOLBENCH(Qin et al., 2023), API -Bank(Li et al., 2023a), and GORILLA API BENCH(Patil et al., 2023) focus on text-only scenario, where the model must map a prompt with text-only tokens to a corresponding function-calls in various languages. More recent efforts like BFCL(Patil et al., 2025) and $\tau$ -Bench(Yao et al., 2024; Barres et al., 2025) broaden the scope to include multi-turn and multi-step tool-use with intermediate user interactions. These datasets not-only are text-only but also assume that the user query is well-formatted text (e.g., no repeating or filler words) and therefore fail to expose the perception-induced reasoning errors that arise in multimodal agents.

MFCL builds-upon and extends by (i) adding *audio* both as trascribed, and native-audio, (ii) includes *vision* modalities along with variations, and (iii) introduces perturbations that highlight failures at the boudries of perception, grounding, and formatting.

**Multimodal Benchmarks:** Large multimodal models are commonly evaluated on image or video understanding suites such as MMBENCH(Liu et al., 2024), SEED -Bench(Li et al., 2023b), MME(Fu et al., 2023), and MMMU(Yue et al., 2024), and agentic benchmarks such as OS-World Xie et al. (2024). While comprehensive in task diversity, these datasets emphasize factuality, and instruction-following. Consequently, they cannot measure whether a model preserves argument types, handles out-of-vocabulary entities, or follows a prescribed schema. MFCL is the first solution to address this gap by enforcing the evaluation of the exact match at both the function and the argument level.

In summary, existing benchmarks either (a) evaluate tool use in text-only settings or (b) test multimodal understanding without enforcing structured outputs. MFCL is the first to unify these threads and provides a single framework to study *multimodal function calling* under controlled perturbations, enabling fine-grained diagnosis and principled progress towards reliable, tool-augmented agents.

## 3 DATA CURATION

This section details the construction of three complementary datasets: **MFCL True Audio**, **MFCL Text Audio**, and **MFCL Vision**. Together they enable a holistic evaluation of multimodal assistant agents across speech, text, and vision.

### 3.1 MFCL TRUE AUDIO

MFCL True Audio converts the original textual queries into realistic spoken utterances through a four–stage process: (1) *natural paraphrasing*, (2) *controllable speech-noise injection*, (3) *synthetic speech generation*, and (4) *real-world acoustic augmentation*. We describe each below.

**Natural Paraphrasing:** To include a robust and diverse set of prompts, we start with text-only function calling queries (Patil et al. (2025)), including single-turn and multi-turn curated by experts and community-contributed. To adapt it to natural language and similar to colloquial conversations, we filter out all queries that contain special characters or symbols. We then rewrite each user query into a more natural conversational-style tone. This is critical, and we discuss this below, for example:

*Original:* "I need to send a letter to Liam Neeson. Find his contact information."

*MFCL processed text:* "Um, can you get Liam Neeson—that's L-I-A-M N-E-E-S-O-N—Liam Neeson's contact info so I can send him a letter?"

**Generating Speech Transcripts:** Spontaneous speech is rife with disfluencies—*filled pauses* ("um," "uh"), *word repetitions* ("the the station"), *hesitations*, *elongations*, and *mid-sentence restarts* ("I want— I mean, we should..."). We broaden this set to include *self-corrections* ("...no, sorry, I meant ..."), *false starts* ("Hey, could you—uh, can you..."), *casual contractions* ("gonna," "wanna"), *conversational markers* ("you know," "I mean"), and explicit *spelling or symbol pronunciations* (e.g., "Contact the admin at j–o-h-n at gmail dot com."). Structural edits such as *preposition dropping* and *sentence restructuring* further mimic everyday speech.

Balancing realism and intelligibility is non-trivial: too many disfluencies can swamp short queries, while too few leave them unnaturally pristine. We therefore partition the transformations into mutually exclusive classes and allow the controller to pick at most one from each class. The controller also receives the full function signature of any downstream tool call, enabling it to veto changes that would corrupt critical arguments. Appendix H provides an overview of the pipeline.

**Synthetic Speech Generation:** The disfluency-augmented text is rendered into waveform audio using a heterogeneous pool of neural TTS services. For each utterance we draw a random combination of speaker identity, accent, speaking rate, and prosody so that no single timbre or vendor dominates.
**Real-World Acoustic Augmentation:** Although TTS output is diverse, it is still unrealistically clean. We therefore apply four augmentation modules that emulate the degradations common in

real deployments (Figure 2). **Additive Environmental Noise Background:** audio from the MU-SAN corpus(Snyder et al., 2015) and CHiME-5 archive(Barker et al., 2018) is mixed at random signal-to-noise ratios (SNRs). **Competing Speech (Double-Talk):** The target utterance is overlaid with interfering speakers sampled from conversational corpora at varied SNRs, assessing robustness in acoustically dense environments like open-plan offices or public transport. **Network Impairments:** Packet loss, jitter, and truncation are synthesized to stress models operating over VoIP or cellular links. We randomly drop 10–40 ms frames, micro-loop short segments, or truncate the tail of the utterance to emulate latency and test the model's ability to act on incomplete information or seek clarification. **Device and Room Effects:** Room impulse responses add reverberation; non-linear transfer curves simulate clipping; random gain shifts emulate speaker-to-microphone distance changes; and short bursts of rubbing or cable noise model mechanical interference.

The parameters of each module are tuned so that queries remain intelligible to robust systems while exposing brittleness in fragile pipelines.

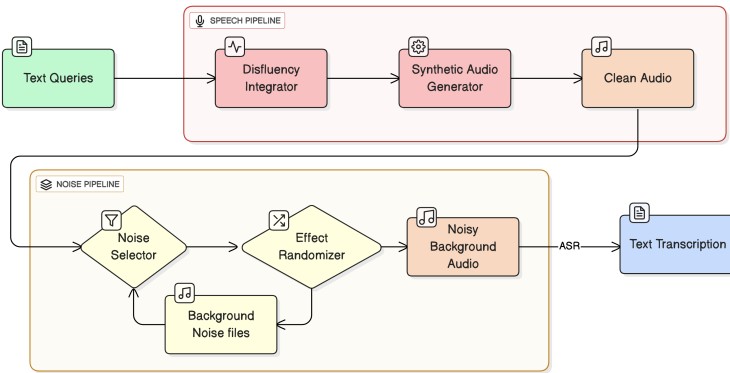

Figure 2: Dual-stage audio augmentation: clean TTS speech is mixed with transformed background noise to create realistic noisy audio for MFCL True Audio; transcripts feed MFCL Text Audio.

## 3.2 MFCL TEXT AUDIO

MFCL Text Audio builds on top of MFCL True Audio by transcribing each synthetic utterance with three distinct ASR engines, exposing systematic ASR variability. For example:

· **Generator 1:** "Um, can you get Liam Neeson—that's L-I-A-M N-E-E-S-O-N—Liam Neeson's contact info so I can send him a letter?"

· **Generator 2:** "Um, can you get Liam Neeson, that's L-I-A-M N-E-E-S-O-N, Liam Neeson's contact info so I can send him a letter?"

· **Generator 3:** "Can you get Liam *Neeeson*? That's L-I-A-M N-E-E-S-O-N, Liam *Neeeson*'s contact info so I can send him a letter?"

MFCL exploits these divergences to overcome the brittleness of single ASR, and thereby provides a comprehensive and robust end-to-end view.

## 3.3 MFCL VISION

The MFCL Vision dataset comprises 250 image-query-trace triplets spanning five image domains (*Places*, *Events*, *Media*, *Sports*, *Shopping*) and five query types (*Locate*, *Temporal*, *Select*, *Identify*, *Quantify*) (Figure 17). Each entry is deliberately constructed such that solving it requires both visual grounding and external web search. See Appendix E

We create the dataset based on the following principles. **Salient visual hints:** Each image contains at least one clear visual clue (defined in Appendix C) the model can leverage through web search. By making the hints accessible, our benchmark prioritizes each model's unobstructed visual reasoning capabilities. **Require tool use:** Each query is written to demand information outside the model's

prior knowledge or baseline visual reasoning; correctly answering requires some web search. **Solvability:** Each image-query pair is constructed such that a non-expert human with access to web search tools could accurately answer the question. **Dependent on external evidence:** Queries are designed such that the answer cannot be derived without consulting up-to-date external sources. They also should not collapse into pure visual reasoning tasks (e.g., OCR, object recognition, or simple counting). **Dependent on the image context:** The image provides essential disambiguation. Without it, the query is unanswerable (e.g., asking "Who owns this team?" without showing a logo). This ensures that both visual understanding and query specifications must interact to produce the final answer.

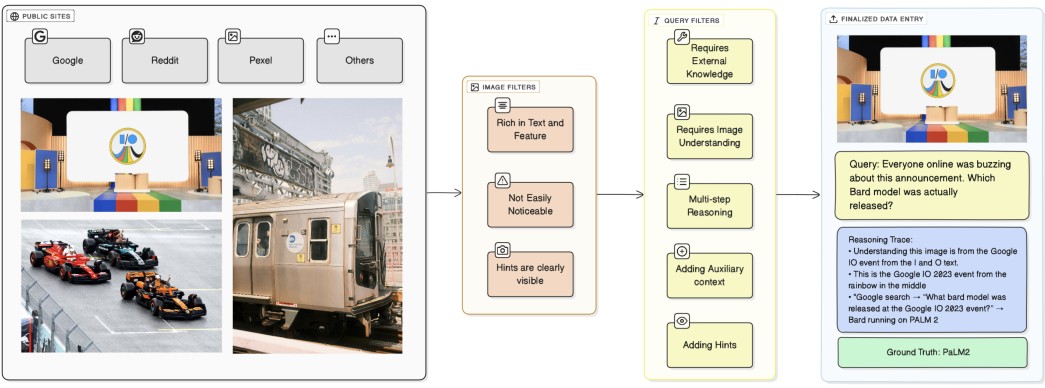

Figure 3: Data construction pipeline for MFCL Vision: curate public-source images (Google, Reddit, Pexels, etc), filter for salient cues, author queries requiring external knowledge, and multi-step visual reasoning with optional context/hints. All queries are solvable by a human.

## 4 EVALUATION SETUP

**MFCL Text Audio**   MFCL Text Audio evaluates (i) correct function calling and (ii) robustness to audio artifacts. **Turn semantics:** Unlike BFCL Text, where non-tool messages end a turn, spoken interaction often requires confirming spellings or values due to homophones and ASR errors. To avoid penalizing necessary clarifications, we allow specific confirmation turns (validated against a dictionary) without ending the interaction. A system prompt informs the model of the audio setting (Appendix J). **Clarification mechanism:** An LLM judge validates clarification requests; if valid, a simulated user provides a brief, whitelisted reply (e.g., spelling). Chitchat is ignored (Appendix I). **Scoring:** Scoring follows BFCL Text: AST matching for single-turn tasks and state-/response-based checks for multi-turn tasks. Clarification turns do not contribute to the final score; they only enable subsequent correct actions.

**MFCL True Audio**   Same as MFCL Text Audio, but with simulated synthetic audio replies.

**MFCL Vision**   Models receive a system prompt defining the output format (Appendix K). We compute exact match on the lower-cased, punctuation-stripped `answer` field to avoid spurious positives from verbose outputs.

## 5 EVALUATION RESULT AND ERROR ANALYSIS

In this section, we analyze the failure modes of models for both audio and vision modalities.

## 5.1 MFCL Audio Failure Modes

| Model | Overall | Expert Curated | | | | Community Sourced | | | | Multi-Turn | | | Hallucination Measure | |
|---|---|---|---|---|---|---|---|---|---|---|---|---|---|---|
| | | Simple | Multiple | Parallel | Parallel Multiple | Simple | Multiple | Parallel | Parallel Multiple | Base | Miss Func | Miss Param | Relevance | Irrelevance |
| GPT-4o-audio-2025-06-03 (Clean Audio) | 60.4 ± 7.1 | 58.6 ± 4.8 | 86.9 ± 5.0 | 80.5 ± 5.9 | 76.0 ± 6.0 | 61.3 ± 6.3 | 67.4 ± 2.8 | 64.3 ± 28.0 | 45.5 ± 21.2 | 37.5 ± 6.5 | 47.5 ± 7.0 | 37.5 ± 6.5 | 72.2 ± 25.1 | 85.4 ± 2.2 |
| GPT-4o-audio-2025-06-03 (Text) | 58.6 ± 6.8 | 51.1 ± 4.8 | 87.4 ± 4.6 | 77.5 ± 5.9 | 74.5 ± 6.0 | 58.4 ± 6.3 | 66.4 ± 2.9 | 50.0 ± 28.6 | 50.0 ± 22.7 | 38.0 ± 6.5 | 44.0 ± 7.0 | 34.5 ± 6.5 | 77.8 ± 22.2 | 84.1 ± 2.2 |
| Qwen3-Omni-Flash-2025-09-15 (Clean Audio) | 55.8 ± 7.0 | 58.3 ± 4.8 | 84.9 ± 5.3 | 77.5 ± 6.0 | 74.0 ± 6.3 | 60.7 ± 6.2 | 58.5 ± 2.9 | 53.6 ± 31.0 | 50.1 ± 21.4 | 26.5 ± 5.8 | 33.3 ± 6.3 | 25.5 ± 5.8 | 61.1 ± 23.7 | 87.5 ± 2.1 |
| GLM-4.5 (Text) | 55.2 ± 7.0 | 47.6 ± 5.0 | 87.9 ± 4.5 | 77.5 ± 5.9 | 66.5 ± 6.5 | 60.9 ± 6.4 | 66.4 ± 2.9 | 71.4 ± 25.2 | 59.1 ± 21.9 | 27.5 ± 6.0 | 22.5 ± 6.0 | 25.5 ± 6.0 | 61.1 ± 25.8 | 86.7 ± 2.1 |
| Qwen3-Omni-Flash-2025-09-15 (Text) | 55.1 ± 6.9 | 52.4 ± 4.9 | 84.2 ± 5.1 | 74.3 ± 6.2 | 72.5 ± 6.3 | 60.7 ± 6.2 | 60.4 ± 3.0 | 53.6 ± 27.5 | 56.8 ± 21.7 | 25.8 ± 5.8 | 31.8 ± 6.3 | 24.0 ± 5.8 | 66.7 ± 24.5 | 87.6 ± 2.0 |
| Claude-Opus-4.1-20250805 (Text) | 53.5 ± 7.0 | 54.5 ± 4.9 | 90.0 ± 4.5 | 73.0 ± 6.0 | 60.5 ± 7.0 | 63.0 ± 6.0 | 69.3 ± 2.8 | 78.6 ± 21.4 | 50.0 ± 22.7 | 25.0 ± 6.0 | 17.5 ± 5.5 | 17.0 ± 5.5 | 61.1 ± 25.8 | 81.8 ± 2.4 |
| Gemini-2.5-Flash (Clean Audio) | 53.0 ± 7.0 | 56.8 ± 4.9 | 83.4 ± 5.5 | 77.0 ± 6.0 | 64.5 ± 6.5 | 61.3 ± 6.3 | 63.8 ± 2.9 | 71.4 ± 25.2 | 50.0 ± 22.7 | 16.0 ± 5.0 | 12.5 ± 4.5 | 13.0 ± 4.5 | 66.7 ± 21.5 | 91.5 ± 1.7 |
| Gemini-2.5-Pro (Text) | 51.5 ± 7.0 | 53.6 ± 5.0 | 80.9 ± 5.5 | 71.0 ± 6.5 | 70.5 ± 6.5 | 63.0 ± 6.0 | 54.3 ± 3.0 | 57.1 ± 26.3 | 63.6 ± 20.7 | 13.5 ± 5.0 | 19.5 ± 5.5 | 13.5 ± 5.0 | 55.6 ± 26.8 | 91.1 ± 1.7 |
| xLAM-2-70b-fc-r (Text) | 51.2 ± 7.0 | 54.1 ± 5.0 | 87.4 ± 4.6 | 75.5 ± 6.0 | 59.5 ± 7.0 | 48.6 ± 6.4 | 52.8 ± 3.1 | 42.9 ± 34.0 | 40.9 ± 21.9 | 28.0 ± 6.0 | 28.0 ± 6.0 | 23.5 ± 6.0 | 77.8 ± 22.2 | 80.1 ± 2.4 |
| Gemini-2.5-Pro (Clean Audio) | 51.1 ± 6.9 | 57.9 ± 4.8 | 82.9 ± 5.5 | 74.5 ± 6.0 | 72.0 ± 6.0 | 60.1 ± 6.0 | 49.5 ± 3.0 | 42.9 ± 34.0 | 54.6 ± 21.6 | 15.5 ± 5.0 | 19.0 ± 5.5 | 13.5 ± 5.0 | 50.0 ± 22.2 | 89.6 ± 1.9 |
| GPT-4o-audio-2025-06-03 (Noisy Audio) | 50.9 ± 7.2 | 43.1 ± 5.0 | 72.9 ± 6.6 | 62.5 ± 6.5 | 62.5 ± 6.5 | 46.5 ± 6.4 | 47.7 ± 3.0 | 35.7 ± 28.0 | 31.8 ± 22.3 | 28.5 ± 6.4 | 39.5 ± 7.0 | 35.0 ± 6.5 | 50.0 ± 22.2 | 85.7 ± 2.2 |
| Grok-4-0709 (Text) | 50.2 ± 7.1 | 53.8 ± 4.9 | 83.4 ± 5.5 | 76.0 ± 6.0 | 67.5 ± 6.9 | 58.9 ± 6.4 | 66.5 ± 2.9 | 71.4 ± 25.2 | 59.1 ± 21.9 | 8.0 ± 4.0 | 3.5 ± 2.5 | 9.0 ± 4.0 | 72.2 ± 25.1 | 84.1 ± 2.2 |
| Qwen3-235B-A22B-Instruct-2507 (Text) | 50.1 ± 7.2 | 46.6 ± 5.0 | 86.4 ± 5.0 | 74.0 ± 6.0 | 68.0 ± 6.9 | 53.9 ± 6.4 | 60.9 ± 3.0 | 64.3 ± 28.0 | 59.1 ± 21.9 | 12.0 ± 4.5 | 8.5 ± 4.0 | 7.5 ± 4.0 | 72.2 ± 25.1 | 88.4 ± 2.0 |
| Qwen3-Omni-Flash-2025-09-15 (Noisy Audio) | 48.2 ± 7.2 | 42.1 ± 5.0 | 71.9 ± 6.6 | 60.0 ± 6.9 | 62.5 ± 6.5 | 48.4 ± 6.4 | 43.8 ± 3.0 | 35.7 ± 28.0 | 40.9 ± 22.5 | 21.0 ± 5.7 | 31.3 ± 6.5 | 26.0 ± 6.0 | 50.0 ± 22.2 | 86.1 ± 2.2 |
| GPT-4o-mini-audio-2024-12-17 (Clean Audio) | 47.2 ± 7.1 | 51.6 ± 4.8 | 80.4 ± 5.5 | 74.0 ± 6.0 | 67.0 ± 6.9 | 58.4 ± 6.3 | 63.4 ± 2.9 | 71.4 ± 25.2 | 36.4 ± 20.7 | 1.5 ± 2.0 | 1.0 ± 1.5 | 2.0 ± 2.0 | 77.8 ± 22.2 | 82.4 ± 2.3 |
| Gemini-2.5-Flash (Noisy Audio) | 45.6 ± 6.9 | 45.4 ± 5.0 | 72.4 ± 6.6 | 57.0 ± 7.3 | 53.5 ± 7.0 | 49.8 ± 6.4 | 46.4 ± 3.0 | 57.1 ± 26.3 | 50.0 ± 22.7 | 15.5 ± 5.0 | 11.5 ± 4.5 | 9.5 ± 4.5 | 33.3 ± 21.5 | 91.0 ± 1.8 |
| Gemini-2.5-Pro (Noisy Audio) | 44.1 ± 7.0 | 41.0 ± 5.0 | 70.9 ± 6.6 | 57.5 ± 7.2 | 62.5 ± 6.5 | 50.2 ± 6.4 | 39.8 ± 2.9 | 35.7 ± 28.0 | 50.0 ± 22.7 | 13.5 ± 5.0 | 23.0 ± 6.0 | 17.0 ± 5.5 | 50.0 ± 22.2 | 86.4 ± 2.1 |
| GPT-5-2025-08-07 (Text) | 41.0 ± 7.0 | 32.1 ± 4.8 | 53.3 ± 7.1 | 55.0 ± 6.8 | 48.5 ± 7.0 | 36.2 ± 6.4 | 39.5 ± 2.9 | 50.0 ± 28.6 | 45.5 ± 21.2 | 11.0 ± 4.5 | 6.5 ± 3.5 | 5.5 ± 3.5 | 44.4 ± 26.8 | 95.0 ± 1.3 |
| GPT-4o-mini-audio-2024-12-17 (Noisy Audio) | 40.3 ± 6.6 | 36.5 ± 4.7 | 66.3 ± 6.6 | 53.5 ± 7.0 | 52.0 ± 7.0 | 42.0 ± 6.3 | 42.3 ± 3.0 | 50.0 ± 28.6 | 22.7 ± 17.9 | 10.5 ± 4.5 | 3.0 ± 2.5 | 4.0 ± 3.0 | 66.7 ± 21.5 | 84.1 ± 2.2 |

Table 1: Model performance on MFCL Text Audio and MFCL True Audio. Evaluation settings: Text = transcribed input, Clean Audio = speech without background noise, Noisy Audio = speech with background noise. End-to-end models on clean audio typically surpass pipelined text-only systems by leveraging contextual and prosodic cues, but their advantage is brittle: noise induces sharp degradation, often below the steadier pipelined baselines—highlighting a trade-off between peak accuracy and robustness.

Background noise sharply degrades every end-to-end (E2E) model we tested (Table 1). Even the strongest system—GPT-4o-audio—loses **9.5 %**. Under noise, the dominant error source shifts from *detecting a request* to *transcribing its details*, producing subtle yet harmful *semantic errors*: the assistant executes an action, but the wrong one.

We observe six recurrent failure modes (FMs):

**FM 1: Intent Blending** Background speech merges with the user's command (e.g., a coworker says "cancel the meeting," which contaminates a flight-booking request).

**FM 2: Premature Execution** A sudden loud sound (door slam, siren) is misheard as end-of-utterance, prompting action on an incomplete command.

**FM 3: Parameter Distortion** Microphone artifacts (clipping, rubbing) mutate key tokens; "fifty" becomes "fifteen," yielding a valid but incorrect function call.

**FM 4: Clarification Misfires** The model requests irrelevant clarifications that's not on named entities, especially in pipelined models with noisy ASR (Fig. 5). ASR transcriptions add noise, creating more uncertainty leading to irrelevant clarifications. E2E performs better due to lesser uncertainty with integrated audio-language understanding.

**FM 5: Conversational Drift** Instead of emitting the required function call, the model slips into a conversational response. RLHF-tuned E2E models default to helpful dialog when uncertainty spikes, trading schema compliance for user-friendly chatter. (Fig. 5)

**FM 6: Named-Entity Errors** Mis-transcribed names, places, or addresses derail argument generation, and the model never asks for confirmation, instead generating a faulty call (Fig. 5). Noisy audio worsens this across all architectures, but is most acute for noisy E2E runs. Specifically for pipelined models, the ASR stage distorts the specific keywords of the transcript despite producing grammatically plausible flow, leading to propagating errors, as the following example shows.

· *Original Query:* "Can you book a flight to Austin for tomorrow morning?"

· *Audio Condition:* True audio with medium-density cafe background noise.

· *ASR Transcript (Clean Audio):* "Can you book a flight to **Austin** for tomorrow morning?"

· *ASR Transcript (Noisy Audio):* "Can you book a flight to **Boston** for tomorrow morning?"

· *Resulting Action:* LLM correctly identified the intent to `book_flight` but received the incorrect entity (`city=Boston`), resulting in a task failure.

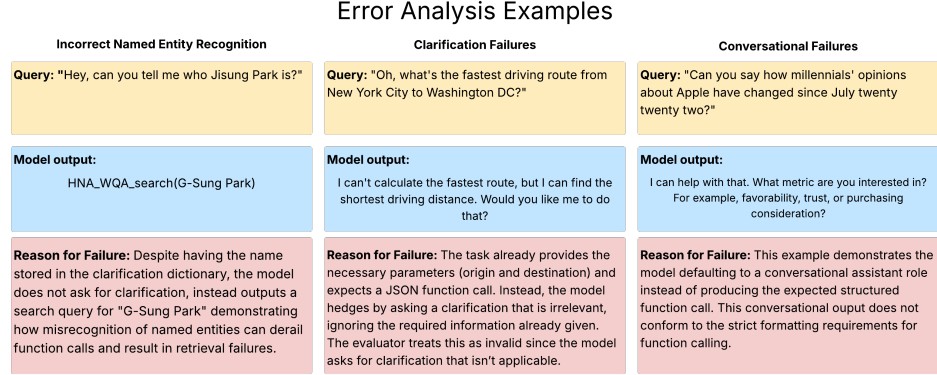

Figure 4: Error Analysis Examples

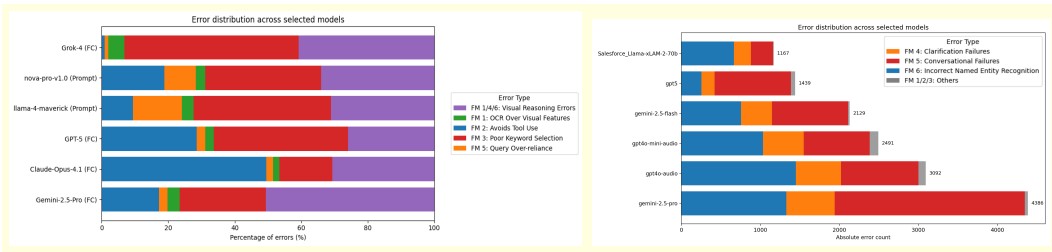

Figure 5: Distribution of failure modes in MFCL Vision (left) and MFCL True Audio (right). See Sections 5.2 and 5.1 for definitions. In the vision plot, modes 4 and 6 are subsumed under *visual reasoning* because both arise from early perceptual missteps that either lock attention onto an irrelevant region or prematurely discard informative cues, ultimately derailing the reasoning trajectory.

## 5.2 MFCL VISION FAILURE MODES

| Model | Overall | Base | Crop16:9 | Crop4:3 | Resize16:9 | Resize4:3 | B&W | Edge | Red&Green |
|---|---|---|---|---|---|---|---|---|---|
| GPT-5-2025-08-07 (FC) | **29.3 ± 5.7** | **34.7 ± 5.9** | **31.9 ± 5.9** | **31.1 ± 5.8** | **30.7 ± 5.8** | **32.7 ± 6.2** | **27.1 ± 5.4** | **17.1 ± 4.7** | **29.1 ± 5.8** |
| Gemini-2.5-Pro (FC) | 26.6 ± 5.5 | 29.9 ± 5.8 | 31.1 ± 5.8 | 28.7 ± 5.8 | 25.9 ± 5.4 | 29.5 ± 5.8 | 25.5 ± 5.4 | 14.3 ± 4.3 | 27.9 ± 5.4 |
| Gemini-2.5-Flash (FC) | 23.1 ± 5.4 | 25.5 ± 5.4 | 26.7 ± 5.4 | 23.1 ± 5.4 | 25.5 ± 5.4 | 24.3 ± 5.4 | 23.1 ± 5.4 | 12.3 ± 4.3 | 23.9 ± 5.4 |
| Grok-4-0709 (FC) | 22.7 ± 5.4 | 25.1 ± 5.4 | 25.1 ± 5.4 | 25.5 ± 5.4 | 21.1 ± 5.1 | 25.5 ± 5.4 | 22.3 ± 5.4 | 11.6 ± 3.9 | 25.1 ± 5.4 |
| o4-mini-2025-04-16 (FC) | 20.0 ± 4.9 | 23.1 ± 5.4 | 22.7 ± 5.4 | 23.5 ± 5.4 | 18.7 ± 5.1 | 20.7 ± 5.1 | 19.9 ± 5.1 | 11.2 ± 3.9 | 20.3 ± 5.1 |
| Qwen3-Omni-Flash-2025-09-15 (FC) | 16.3 ± 4.7 | 16.5 ± 4.8 | 18.0 ± 4.8 | 19.5 ± 5.0 | 16.9 ± 4.7 | 16.3 ± 4.7 | 15.5 ± 4.7 | 11.6 ± 3.9 | 15.4 ± 4.6 |
| Claude-Opus-4-1-20250805 (FC) | 15.9 ± 4.6 | 16.7 ± 4.7 | 18.3 ± 5.1 | 17.5 ± 4.7 | 15.9 ± 4.7 | 17.5 ± 4.7 | 15.5 ± 4.7 | 11.6 ± 3.9 | 13.9 ± 4.3 |
| Claude-Sonnet-4-20250514 (FC) a | 14.9 ± 4.5 | 16.7 ± 4.7 | 17.5 ± 4.7 | 17.9 ± 5.1 | 13.9 ± 4.3 | 18.3 ± 5.1 | 12.3 ± 4.3 | 8.4 ± 3.6 | 14.3 ± 4.3 |
| GPT-4o-2024-11-20 (FC) | 11.7 ± 4.0 | 11.9 ± 4.3 | 14.3 ± 4.3 | 15.5 ± 4.7 | 10.4 ± 3.9 | 12.8 ± 4.3 | 11.9 ± 4.3 | 4.8 ± 2.8 | 11.6 ± 3.9 |
| Llama-4-Maverick-17B-128E-Instruct-FP8 (FC) | 10.6 ± 3.8 | 12.8 ± 4.3 | 11.2 ± 3.9 | 9.6 ± 3.9 | 10.8 ± 3.9 | 12.3 ± 4.3 | 11.6 ± 3.9 | 6.8 ± 3.2 | 10.0 ± 3.9 |
| Amazon-Nova-Pro-v1:0 (FC) | 10.1 ± 4.0 | 12.8 ± 4.3 | 10.4 ± 3.9 | 9.6 ± 3.9 | 9.6 ± 3.9 | 10.4 ± 3.9 | 9.6 ± 3.9 | 6.8 ± 3.2 | 11.6 ± 3.9 |
| GPT-4o-mini-2024-07-18 (FC) | 9.0 ± 3.4 | 10.0 ± 3.9 | 11.2 ± 3.9 | 8.4 ± 3.6 | 10.0 ± 3.9 | 9.2 ± 3.5 | 9.2 ± 3.5 | 6.4 ± 3.2 | 8.0 ± 3.6 |
| Mistral-Medium-2508 (FC) | 8.7 ± 3.5 | 10.4 ± 3.9 | 10.0 ± 3.9 | 8.4 ± 3.6 | 10.8 ± 3.9 | 11.2 ± 3.9 | 9.2 ± 3.5 | 1.2 ± 1.6 | 8.0 ± 3.6 |
| Pixtral-Large-2411 (Prompt) | 8.4 ± 3.6 | 9.6 ± 3.9 | 10.0 ± 3.9 | 7.6 ± 3.6 | 8.8 ± 3.5 | 11.2 ± 3.9 | 6.0 ± 3.2 | 6.0 ± 3.2 | 8.0 ± 3.6 |
| GLM-4.5V (Prompt) | 7.9 ± 3.5 | 10.0 ± 3.9 | 5.2 ± 2.8 | 9.2 ± 3.5 | 7.2 ± 3.2 | 10.4 ± 3.9 | 8.8 ± 3.5 | 3.6 ± 2.4 | 8.8 ± 3.5 |
| Command-A-Vision-07-2025 (Prompt) | 6.2 ± 3.1 | 6.8 ± 3.2 | 6.0 ± 3.2 | 6.0 ± 3.2 | 7.2 ± 3.2 | 6.0 ± 3.2 | 7.2 ± 3.2 | 4.4 ± 2.8 | 6.0 ± 3.2 |

Table 2: Model performance on MFCL Vision across all 8 variations. FC means native function calling support, Prompt means prompt-based walkaround. Base means no augmentation was applied to the image. Edge detection variant brings the biggest drop in performance (almost halved). All other variations have similar performance.

We run the following ablation studies on MFCL Vision: *(i) color manipulations*, where we convert images to black-and-white (B&W) or remove the blue channel (Red&Green only); *(ii) edge-based transformations*, where we apply standard edge detection; and *(iii) aspect-ratio changes*, where we either crop or resize images to 4:3 or 16:9. For cropping, we ensure that important information remains visible, while resizing preserves all details but can distort shapes.

Our discussion centers on six recurring failure modes, which, taken together, underscore the core obstacles that vision–function calling must overcome to achieve reliable performance.

**FM 1A: Visual Reasoning Errors**   Failures largely stemmed from the known weaknesses of VLMs that were deliberately targeted during image curation and tuning. Models struggle to balance textual with visual information. In particular, they show a tendency to concentrate excessively on textual elements, ignoring other salient visual cues in the image. (Appendix F)

**FM 1B: Ignored Image Text**   Models ignore on-image text, defaulting to generic descriptions and underperforming on tasks that require text extraction (e.g., seeding search). (Appendix D Figure 16)

**FM 1C: Subset Confusion**   Faced with many look-alike objects models often misidentify subsets. In overloaded scenes they tend to omit items or hallucinate extras, skewing keywords and basic counting. ( Figure 6)

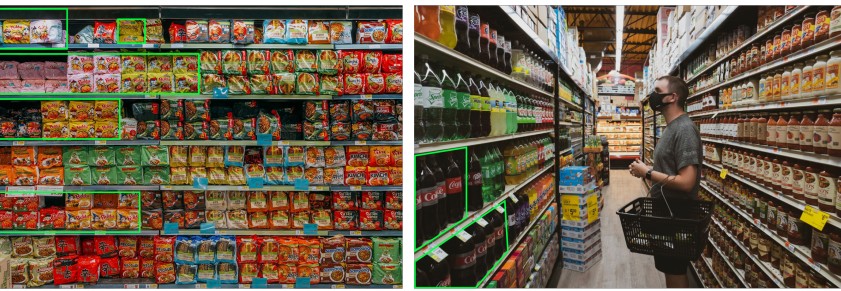

Figure 6: Subset confusion examples. Green boxes mark target subsets, but models often hallucinate extra items or miss targets, especially in the *Shopping* category.

**FM 1D: Myopia**   In deep-focus images, models over-attend to salient foreground objects at the expense of more relevant background cues—even when named in the query (Figure 7). This foreground bias skews search, with distractors driving keyword selection.

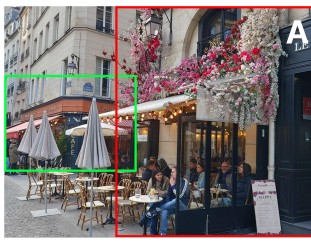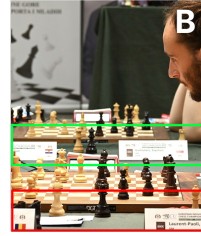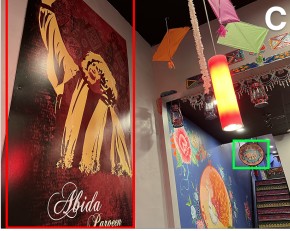

Figure 7: Example entries exhibiting depth-induced bias. The green box indicates the key clue and the red box highlights the foreground distractor. Cases A and B had associated queries that explicitly informed the model to reason on something further back in the image (e.g. referring to the *"cafe with orange signage"* or *"chessboard in the back"*). In all cases, the foreground distractor significantly confounded reasoning across most models.

**FM 2: Avoiding Tool Use**   Sometimes models avoid searching when uncertain. Instead, they either ask clarifying questions (Figure 8) or provide best-guess answers based solely on internal reasoning despite having access to search tools (Figure 9).

**FM 3: Poor Keyword Selection**   The model generates vague or irrelevant queries. Rather than extracting specific visual details (e.g., text, symbols, spatial cues) into targeted keywords, it defaults to generic phrases (Figure 10), yielding irrelevant search results.

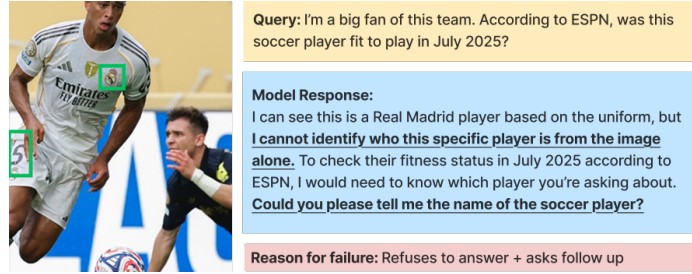

Figure 8: This example displays the player's team logo and jersey number, which are major clues the models should recognize before making any tool calls. Despite this, the model stalled and requested more input. We sent back a user reply: *'I don't know'*, but the model still refuses.

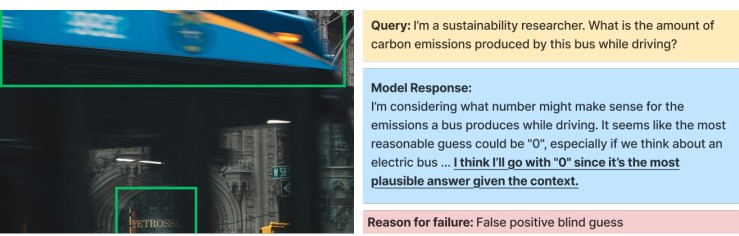

Figure 9: Rather than reasoning about the image or leveraging search, the model simply makes a blind guess, offering meta-reasoning about what might be a "plausible" answer given the query text.

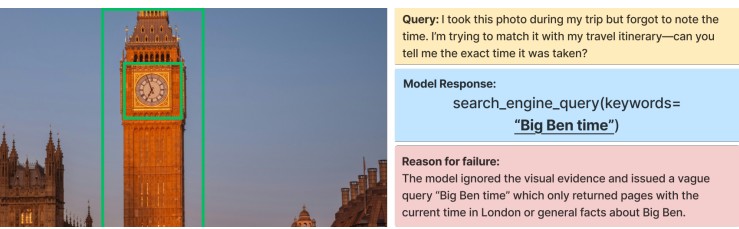

Figure 10: This example implicates analog clock reading and spatial orientation relative to the sun to determine the exact time. The model failed to combine any of this nuance to produce specific keywords for search.

**FM 4: First-hop Bias**   Models latch onto an initial assumption and resist updating, even when faced with contradictory evidence (sometimes rationalized away as *"misremembering"*, Figure 11). In other cases, it reasons correctly during the trace, but the final output reverts to the initial incorrect guess, discarding its own reasoning.

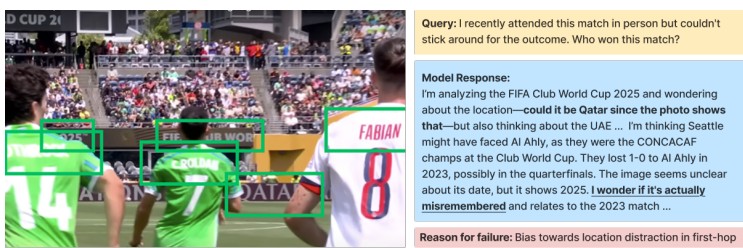

Figure 11: Example of first-hop bias. Even with clear visual evidence of teams, players, and year, the model fixates on a Qatar Airways ad during its first reasoning step. This anchors it to an incorrect path and leads it to dismiss the correct temporal evidence.

**FM 5: Over-reliance on Query Text** Models prioritize keywords from the initial query over visual evidence, issuing generic tool calls (Figure 12) and missing the unique clues in the image.

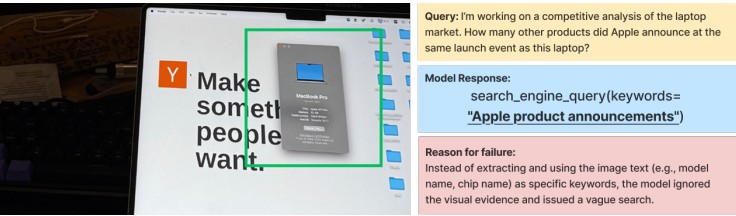

Figure 12: This is an example of query over-reliance. Despite the clear *About This Mac* window, the model ignored the text and issued a vague search based only on the user's query.

**FM 6: Abandoning promising leads** Models note alternative hypotheses but settle on the first plausible answer, often reinforced by search hits, leaving correct options unexamined (Figure 13).

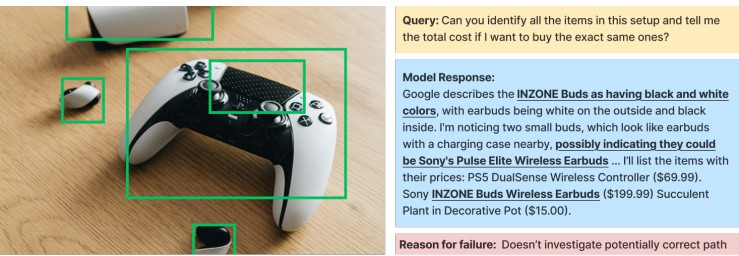

Figure 13: This example entry requires strong object recognition as it demands multiple specific item look-ups. While able to surface potential possible answers, the model latches onto the first plausible hypothesis and ignores the stronger alternative it had already identified.

### 5.2.1 ABLATION ANALYSIS

Examining ablation accuracies (Table 2), the *Edge Detection* variant produces the largest performance drop; all other treatments yield only minor degradation.

**Edge detection removes key cues:** As a subtractive transform, it suppresses contrast and erases fine-grained anchors (e.g., small text, thin logos), depriving models of the visual evidence needed to compose targeted search queries. Performance collapses as a result (Appendix G).

**Color ablations alter accuracy and strategy:** Both *Black-and-White* (B&W) and *Red&Green* (R&G) reduce accuracy, confirming color's discriminative value. B&W can also shift strategy: in Appendix 14 (top), original and R&G yield the same incorrect no-search guess, whereas B&W triggers a search that finds the correct answer; in Appendix 14 (bottom), original and R&G stop at refusal. These patterns suggest that reduced color fidelity acts as an uncertainty cue, nudging models from guess-only behavior toward tool-assisted reasoning.

## 6 CONCLUSION

MFCL provides the first *multi-modal* benchmark for tool-augmented language models, exposes consistent cross-modal failure modes, and offers lightweight, reproducible metrics that enable rapid iteration. Our analysis reveals that current state-of-the-art models still treat tool use as an optional afterthought, especially under noise or visual perturbations, and that simple augmentations such as edge detection can erase decades of accuracy gains. We release MFCL, all code, and evaluation and analysis tools to spur research on robust, tool-aware reasoning.

## ETHICS STATEMENT

Our work introduces MFCL, a multi-modal function-calling evaluation that combines publicly available text, image, and audio data. All source datasets are either (i) released under permissive licenses (e.g., CC-BY, CC-BY-SA) or (ii) in the public domain; we redistribute only metadata and references, never raw copyright-protected content. No human-subject experiments were conducted, and no personally identifiable information is included, so institutional review-board (IRB) approval was not required. We audited MFCL for sensitive attributes (gender, race, religion) and found none representation of sensitive categories; nevertheless, downstream users should be aware that biased function schemas could amplify demographic stereotypes. To mitigate misuse, we provide detailed documentation describing collection, filtering, and annotation rules; a usage license that forbids deploying MFCL to train or evaluate models intended for surveillance, disinformation, or other malicious purposes; and release a standardized bias-and-toxicity evaluation script to help practitioners quantify harmful behaviors. The authors declare no conflicts of interest or third-party sponsorship that could unduly influence the results.

## REPRODUCIBILITY STATEMENT

We have taken several steps to facilitate reproducibility: All preprocessing code, function schemas, and evaluation scripts are avaiable publicly online in an anonymous repository. Section 3 details dataset construction; Appendix A lists every upstream source, license, and filtering criterion. Section 4 and Appendix B describe the metrics and provide pseudocode for each.

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

## A  MFCL VISION: FC MODELS VS PROMPTING MODELS

Our benchmark evaluates three kinds of information that models can use when forming a search query: the text of the user's question, text found in the image, and visual features of the image. The way models utilize these sources differs significantly, creating a noticeable gap between prompt-only and FC model approaches. Prompt-only models often ignore image text and visual cues. Instead, they repeat or mimic the user's question in the search call, with no references to visual elements. For example, the model searched "identify building in image square footage" when asked about a building's size, and maverick searched "season in image" when asked to identify what season is going on in the image. Other typical cases include queries like "blue line pointing to structure in image," "bottle in image," or "store in image espresso price." These examples show that prompt-only models mistakenly assume the tool can inherently "see" the image, which leads to poor search results.

Within prompt-only models, we also observe differences in how effectively they compensate for this limitation. Some models tend to recognize visual features but uses them incorrectly, leading to confident but wrong queries. Other models show better use of image text by weaving it into search queries, however still share the same structural weakness of assuming the tool can "see". In contrast, FC models are explicitly forced to separate and fill arguments for query text, image text, and visual features. This design prevents the "tool sees the image" assumption and leads to more grounded and reliable searches overall.

## B  MFCL VISION: COLOR ABLATION EXAMPLE

### Q: What parade in what year did this balloon design make its debut?

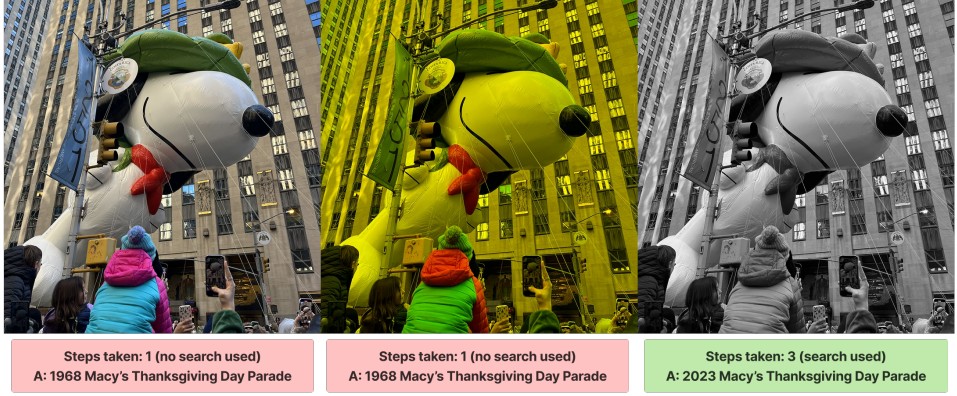

### Q: What total score did this individual receive?

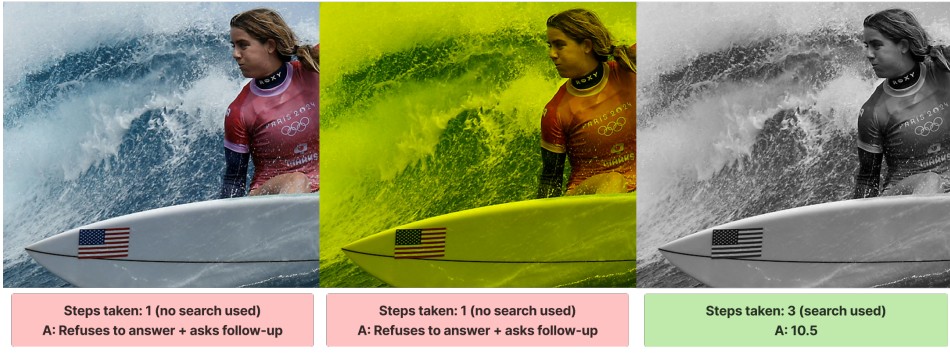

Figure 14: Example entries with color ablations applied. Across both sample entries, original and R&G similarly ignore tools and fail while the B&W counterpart successfully uses search to recover the right answer

## C MFCL VISION: VISUAL CLUE EXAMPLE

We define a visual clue as a non-textual image feature element that uniquely anchors the image to answer the query. These may be obvious features—logos, text, uniforms, product labels—or more subtle ones such as maps, architectural styles, color schemes, event layouts, jerseys, packaging, or the arrangement of objects. Regardless of form, each clue should be sufficiently informative to support the model in identifying the correct scene and grounding its subsequent tool use. The model should ideally begin by detecting one strong clue relevant to the query—such as a sponsor logo on a jersey, a restaurant name on a uniform, or the distinctive shape of a building—which then guides the choice of search terms and the overall interpretation.

Each image category exhibits specific visual features, such as concert venues with recognizable lighting and signage, grocery stores with structured aisles, or sports events with identifiable jerseys and logos. These patterns serve as the model's first step in narrowing down the scene and linking it to the user's query.

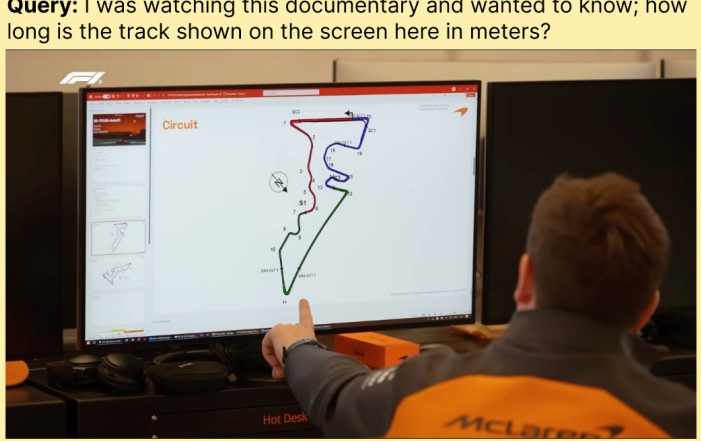

Figure 15: The image provides clear visual evidence of the event via signage in the background and partially complete identifying information of the players and teams via jersey designs and last names. Combining these cues provides a sufficient level of information to construct pointed search queries to approach the answer.

## D MFCL VISION: IS MODEL SIZE THE BOTTLENECK?

One might think that the bottleneck of a model's performance is its training size, due to its ability to recognize certain visual features from images. However, we argue that the bigger limitation is the inability to make an appropriate tool call once a useful clue has been spotted. Images in our dataset are designed not to test raw factual recall but to require the model to notice a clue and then use it effectively in subsequent tool calls.

We also observe a sharp difference in search quality between larger and smaller models. Larger models often manage to identify the clue (e.g., a logo or a sign) and use it properly in their queries. Smaller models, on the other hand, frequently skip the clue entirely and instead fall back to surface-level descriptions of the image (Figure 16). This prevents them from ever reaching the relevant information.

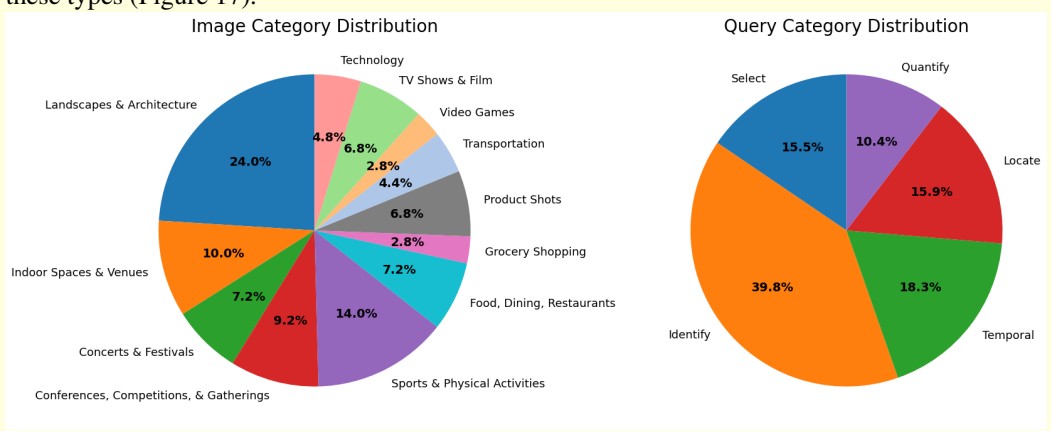

Figure 16: This example shows a woman performing with a large venue logo clearly visible in the background (the "H" inside a circle). Instead of trying to discern the venue from the logo hint, the model performs a poor keyword search using a literal description of the image. This query only led to stock photos and irrelevant results, illustrating how smaller models bypass the reasoning step and never exploit the available clue.

## E    MFCL VISION: CATEGORY DISTRIBUTIONS

To ensure robust evaluation, the MFCL Vision suite intentionally spans a diverse set of image categories and query types. This breadth forces models to ground tool calls using cues from a wide range of visual contexts and reasoning challenges. Below we provide category distributions across these types (Figure 17).

Figure 17: MFCL Vision image and query category distributions.

## F    MFCL VISION: ENTRY TUNING AND VALIDATION

Each triplet undergoes iterative tuning (Figure 18) to calibrate difficulty: adding irrelevant conversational context, enforcing multi-hop chains, injecting lightweight hints, or selectively cropping the image. We retain only those entries that remain solvable by humans but consistently defeat state-of-the-art VLMs.

We commit queries that maintain human solvability but consistently fail on the majority of state-of-the-art models. These include cases of repeated incorrect answers, excessively long reasoning traces, verbose non-answers or expressions of uncertainty, and outright refusals to attempt a solution. Achieving this balance often requires an extensive trial-and-error process of incremental, delicate adjustments, as annotators must tune entries just enough to surface reliable failure patterns without rendering them trivial or unsolvable.

## G    MFCL VISION : EDGE DETECTION

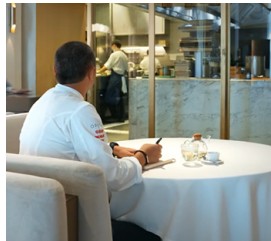

Figure 18: Anatomy of an image-query entry that received several tuning treatments. The query is decomposed into distinct components: orange indicates auxiliary context, blue highlights the actual information request, green provides query hints to constrain the search space, and pink specifies the return format used to facilitate string matching for evaluation.

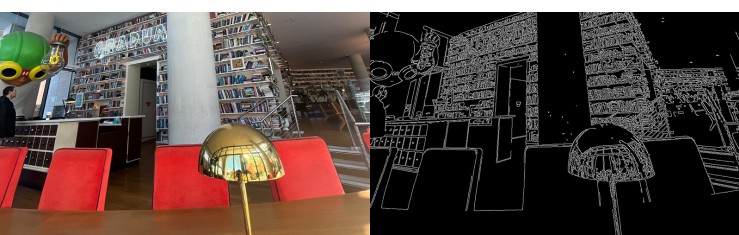

Figure 19: Edge detection ablation example. The accompanying query for this entry reads: *"I was visiting this hotel while attending a conference. Who designed the sculpture to the left?"*. We observe that both color and textual information (e.g., "GRADUA") are completely lost.

## H  SPEECH PIPELINE

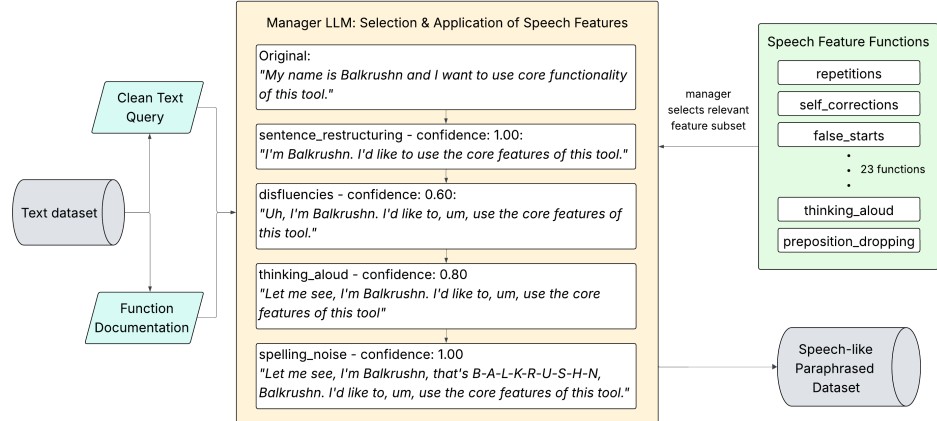

Figure 20: We analyze each query's length and complexity to decide how many speech features to apply, score their relevance, and select the top ones. Each selected feature is an independent function applied sequentially in the chosen order, transforming the query to be natural and speech-like. This enables adaptive and context-sensitive feature injection while preserving semantic meaning.

## I  PROMPT TO THE LLM MODEL FOR JUDGING CLARIFICATION

The LLM model sees the intended request, the ASR text, the assistant's message, and the allowed clarification keys. It approves only if the assistant is explicitly confirming spellings/values that appear in the whitelist. Otherwise, it rejects.

You are a judge for an audio-chat scenario where a user speaks and
    an ASR system transcribes their speech for the assistant. The
    assistant only sees text (the ASR transcript), which is likely
    to contain transcription errors.

You are given:
- intended_request: the user's original, ground-truth intent.
- asr_text: the ASR-transcribed text the assistant saw.
- allowed_clarifications: a set of fields with canonical spellings/
    values the user can confirm (e.g., names, IDs, emails, dates,
    numbers).
- assistant_message: the assistant's latest message.

Your job: decide whether assistant_message is a clarifying question
    specifically about spelling/verification of intent or exact
    strings/values that could plausibly be misheard (e.g., names,
    organizations, emails, serials/IDs, numbers, dates, addresses,
    SKUs). Do not allow general follow-ups (preference, steps to
    proceed, etc.).

Decision rules:
1. Classify the message as a spelling confirmation only if it
    explicitly asks to verify the exact spelling/format/value of one
     or more items (e.g., "Is it Mikaela or Michaela?", "Can you
    spell the email?", "Is the order number A1B-52?").
2. The request must be reasonable given the ASR risk (i.e., the item
     is a proper noun, key value, or easily misheard token relevant
    to the task).
3. To approve (allowed=true), all the topics the assistant asks to
    confirm must be present in allowed_clarifications. If any
    requested item is absent or ambiguous, set allowed=false.
4. Output only a JSON object with two fields:
- allowed: boolean
- message: string (a concise simulated user reply only when allowed=
    true; otherwise empty "").
1. When allowed=true, compose message by supplying only the
    requested values with correct spelling/format from
    allowed_clarifications. Keep it brief (one short sentence or a
    compact list). Do not include extra commentary, JSON, or fields
    the assistant didn't request.
2. If the assistant's message is not a confirmation request, touches
     topics outside spelling/format/intent verification, or requests
     values not available in allowed_clarifications, return allowed=
    false with message="".

Edge cases:
- If the assistant mixes spelling confirmation with unrelated
    questions, treat it as not allowed unless the spelling part
    stands alone and you can fully answer it from
    allowed_clarifications.
- Treat homophones and near-matches as spelling checks (e.g., "Brian
    /Bryan", "Steven/Stephen", letters vs. digits).
- Normalize case/diacritics but preserve canonical spelling in the
    final answer.
- Never reveal intended_request verbatim; only return the specific
    confirmed values.

The user's original intended request is: {the original text mode
    bfcl question}

```
The ASR-transcribed output is: {the transcribed text from the audio,
    which is also the input to the model}

assistant_message: {the model's response}

allowed_clarifications (topic -> answer): {the
    allowed_clarifications}
```

## J  SYSTEM PROMPT FOR MFCL TEXT AUDIO

To inform models that they are in an audio setting, we prepend a short **system prompt** to each conversation:

```
You are a voice assistant that interacts with the user exclusively
    through spoken conversation. You receive user utterances as text
     transcribed by an upstream ASR system and your replies are
    delivered to the user through a TTS system. Follow the rules
    below at all times:

1. Language

* Mirror the user's language. Respond in the same language detected
    in the transcription.

2. Robustness to ASR Errors (Important)

* Although the upstream ASR system is designed to be robust, it may
    still make mistakes.
* Do not trust the transcription text blindly, especially on
    important information. You should assume the transcript may
    contain recognition mistakes.
* If the text appears garbled, double check with the user instead of
     guessing.

3. Clarity for TTS

* When responding to the user, you should **spell out acronyms** as
    separate letters with spaces ("A I M L"), and **chunk long
    numbers** into 2- or 3-digit groups, separated by short pauses (
    "one-two-three, four-five-six").
* Favor spoken-language style: short sentences, everyday vocabulary,
    and natural contractions.
```

## K  SYSTEM PROMPT FOR MFCL VISION

During evaluation, the model receives explicit instructions for response formatting (via system prompt):

```
For your final answer to the user, you must respond in this format:
{'answer': A short and precise answer to the question, 'context': A
    brief explanation of how you arrived
at this answer or why it is correct}.
If you do not know the answer, respond with {'answer': 'I do not
    know', 'context': 'I do not know'}.
```

```
If you think the question cannot be properly answered, response with
    {'answer': 'I cannot answer this
question', 'context': A short reason explaining why this question
    cannot be answered}.
```

