# OpenReview forum: "MFCL: A Multi-modal Function Calling Evaluation for Large Language Models"
_ICLR.cc/2026/Conference — ICLR 2026 Conference Desk Rejected Submission_

### Official Review · Reviewer_wS9C · 2025-10-27

**Soundness:** 3
**Presentation:** 3
**Contribution:** 3
**Rating:** 6
**Confidence:** 3

**Summary:**

1. This paper introduces MFCL, the first unified benchmark to evaluate structured function calling from multi-modal inputs (speech and vision).

2. It systematically injects realistic perception perturbations and uses exact-match automated scoring to diagnose failures

3. Results reveal significant degradation under noise and visual distortions, exposing critical weaknesses such as tool avoidance, keyword selection errors, and conversational drift in modern models.

**Strengths:**

1. Realistic perturbation design that simulates real-world failure conditions in multimodal function calling and thoroughly analyzes their impact.

2. Extensive evaluation across multiple leading commercial models, demonstrating substantial experimental effort and providing meaningful comparative evidence for the community.

**Weaknesses:**

1. Despite arguing for “real-world audio robustness,” the dataset relies on synthetic TTS rather than human-recorded speech.
2. The benchmark’s core metric (exact-match JSON output) misaligns with real agent objectives, overlooking task-level success, semantic equivalence, and cost-aware behaviors.
3. The turn and clarification rules constrain reasonable uncertainty-handling strategies, potentially biasing models toward brittle “just emit JSON” behaviors instead of safe, real-world interaction patterns.

**Questions:**

While the benchmark is valuable, I feel there are several questions:
1. How does strict exact-match scoring avoid misaligning the benchmark with real-world multi-turn agent behavior (uncertainty handling, clarification, self-correction)?
2. The turn semantics and clarification rules only allow spelling/value confirmations, while ignoring broader ambiguity resolution. How do these constraints avoid discouraging realistic uncertainty-handling strategies that agents must perform in practical deployments?
2. Given the recent emergence of audio-based function-calling benchmarks, is it appropriate for MFCL to claim to be the “first” in this space, and could the authors clarify the concrete differences that distinguish MFCL from prior speech-focused evaluations?

---

> ### Author Response · Authors · 2025-11-26
>
> We thank the reviewer for the insightful comments. We address your concerns below.
>
> ---
>
> **Q1. Synthetic Audio vs. Human-Recorded Speech for Real-World Audio Robustness**
>
> Our goal with MFCL True Audio is to approximate the range of conditions faced by deployed voice assistants, not to perfectly reproduce any one conversational corpus. We address realism at the following levels:
>
> - **On the text side**, we did not simply feed clean text to a TTS engine. As detailed in Section 3.1, we utilized a "Manager LLM" to inject 23 distinct types of spontaneous speech phenomena, including filled pauses ("um," "uh"), self-corrections ("no, sorry, I meant..."), and false starts.
> - **On the speech side**, we synthesize audio using a heterogeneous pool of neural TTS voices spanning multiple prosodies, accents, and speaking rates, which avoids overfitting to a single timbre or prosody profile.
> - **For acoustic conditions**, we mix speech with environmental noise and reverberation derived from realistic corpora (e.g., MUSAN and CHiME-5, which consists of real domestic recordings), and simulate channel effects such as packet loss, clipping, and room impulse responses. The SNR ranges and distortion strengths were initially tuned via iterative author listening so that utterances remained clearly intelligible while exhibiting plausible “busy café”, “noisy call”, or “reverberant room” characteristics.
>
> ---
>
> **Q2. Human Evaluation to Confirm Perceptual Naturalness**
>
> To directly verify perceptual naturalness, we conducted a human evaluation on our synthetic audio. We sampled 200 utterances (100 clean, 100 noise-augmented) and recruited 20 raters (internal volunteers). Each rater evaluated a random subset on a 1–5 scale (where 1 is worst and 5 is best) for: **Speech naturalness** (*“Does the voice sound like a plausible human speaker?”*), and **Acoustic realism** (*“Does the background / channel sound like a plausible real-world condition?”*; asked only for noisy clips). The results are shown below.
>
> | Metric | Clean Audio (Mean ± SD) | Noisy Audio (Mean ± SD) |
> | :--- | :---: | :---: |
> | **Speech Naturalness** | 4.1 ± 0.7 | 3.9 ± 0.8 |
> | **Acoustic Realism** | N/A | 3.8 ± 0.7 |
>
> In 83% of noisy clips, raters gave ≥3 on both speech naturalness and acoustic realism.
> These results indicate that listeners generally perceive our synthetic audio as natural and realistic.
>
> ---
>
> **Q3. Alignment of Exact-Match JSON Metric with Real Agent Objectives (Task-Level Success, Semantic Equivalence, and Cost-Aware Behaviors)**
>
> We thank the reviewer for pointing out the need to align with agent objectives like task-level success. In the specific domain of Function Calling, which is the focus of the paper, precise execution is a prerequisite for task success, not a distraction from it. We address the three components of this concern:
>
> **Why Exact-Match Equals Reliability**: In tool-use scenarios, syntax is semantics. APIs are code-bound interfaces. If an API requires `date="2025-07-15"` and the model outputs `date="July 15th"`, the call fails in a production environment regardless of semantic intent. Many benchmarks rely on "brittle LLM judges" to evaluate semantic closeness. By using expert-verified ground truth traces, MFCL removes the noise of auto-graders. We measure the model's ability to trigger a successful execution, rather than a judge's subjective opinion on whether a hallucinated parameter "looks close enough."
>
> **Incorporating Task-Level Success**: MFCL does not ignore task-level goals; we adapt the metric based on the modality:
>
> - **Vision Suite**: Evaluated on the final answer rather than the raw tool call. Models must use visual cues to construct calls that lead to the correct factual answer. This is a purely task-level metric.
>
> - **Audio Suites**: We employ AST (Abstract Syntax Tree) matching (adopted from BFCL). This explicitly handles semantic equivalence (e.g., recognizing that `func(a=1, b=2)` is identical to `func(b=2, a=1)`). Furthermore, for textual answers, we strip punctuation and casing to penalize only genuine argument errors while allowing for harmless stylistic variation.
>
> **Cost-Awareness**: MFCL is designed to support cost-aware evaluation. Our harness measures  tokens used in the eval, allowing developers to calculate the precise price of an evaluation run. However, we deliberately decoupled cost from the primary accuracy score to provide a pure performance metric, (since cost is also a function of economics e.g., new LLM providers running their model at a loss, and varies drastically over time) leaving cost-efficiency tradeoffs to the developer’s specific deployment context.

---

> ### Author Response · Authors · 2025-11-26
>
> **Q4. Clarification Rules and Their Effect on Ambiguity Resolution Strategies**
>
> You raise a valid point: in open-ended deployment, agents should clarify ambiguous intents. However, as a standardized benchmark, MFCL imposes these constraints for three methodological reasons:
>
> **Control of Query Completeness**: Unlike open-ended dialogue, MFCL tasks are expert-verified to be semantically complete. If a user explicitly states "What’s the weather in Berkeley?" and the model asks "Do you mean the city Berkeley in California?", it has failed to extract the provided context. Under our protocol, this is correctly penalized as failure to follow instructions, whereas looser rules would mask this as "realistic uncertainty."
>
> **Isolating Perception vs. Reasoning Errors**: We strictly limit "allowed clarifications" to spelling and value confirmations (e.g., "Did you say 'John' or 'Jon'?") to target the specific uncertainties introduced by our acoustic noise pipeline. We reward agents for recognizing acoustic ambiguity and being cautious while penalizing agents that use clarification as a "crutch" to mask a failure to ground the query.
>
> **Penalizing "Tool Avoidance"**: A key contribution of our paper is identifying failure modes like *Conversational Drift* (FM 5) and *Tool Avoidance* (FM 2). RLHF-tuned models often default to polite chatter ("What would you like me to do?") rather than taking action. Relaxing clarification rules would allow models to stall, hiding the *Clarification Misfires* (FM 4) where models ignore clear evidence.
>
> ---
>
> **Q5. Concrete Differences Between MFCL and Previous Speech-Oriented Benchmarks**
>
> MFCL is the first to comprehensively benchmark the **Text-Model-in-the-Loop** (ASR-LLM-TTS) pipeline for function calling, which reflects the vast majority of current industry deployments for audio agents.
>
> Compared with existing works for speech-focused evaluations:
>
> **CAVA**: CAVA focuses on conversational dynamics (tone, turn-taking, latency). While it touches on function calling, it evaluates only the invocation of the correct function name. **It does not validate parameter values.** In industry settings, correct arguments are critical; calling transfer_money is useless without the correct amount and recipient. MFCL evaluates both.
>
> **AU-Harness**: AU-Harness evaluates "Speech Function Calling" by simply running TTS over text-based function calling queries. And more importantly, **it remains a single-turn evaluation**. MFCL introduces multi-turn interactions and specifically injects acoustic challenges (homophones, background noise) that require the model to engage in clarification loops, a dimension AU-Harness entirely overlooks.

---

### Official Review · Reviewer_ReJp · 2025-10-31

**Soundness:** 1
**Presentation:** 2
**Contribution:** 2
**Rating:** 2
**Confidence:** 3

**Summary:**

The paper addresses two limitations of current multimodal benchmarks: 1) reliance solely on text-based tools, and 2) lack of feedback across different modalities and fine-grained details. To tackle these issues, the authors propose MFCL (Multi-modal Function Calling Evaluation), which consists of three components: True Audio, Text Audio, and Vision. The authors establish specific guidelines for generating each type of data. They evaluate several mainstream models on the proposed benchmark and analyze the results from both audio and visual perspectives. From the audio perspective, they find that current multimodal large models are highly sensitive to speech noise and often fail to confirm critical entities, leading to task failure. From the visual perspective, they observe that these models still lack sufficient attention to details, along with limited tool-calling capabilities and self-correction abilities.

**Strengths:**

There is currently a scarcity of in-depth evaluation benchmarks for multimodal large models, and this paper contributes meaningfully to this field.

**Weaknesses:**

1. The introduction is somewhat disorganized. The authors mention two research gaps, but starting from the third paragraph, they delve into the construction of the benchmark without directly linking it to how these gaps are addressed. I suspect the authors intended to highlight the lack of a benchmark combining API and multimodal evaluation, but the current version is hard to follow, making the motivation unclear. Additionally, Figure 1 is not referenced in the text.

2. There is a lack of data validation, particularly human evaluation, making it difficult to assess the benchmark’s quality and potential biases. Furthermore, certain details remain unclear. For instance, the authors state that vision data requires "one clear visual clue," but there is no in-depth analysis of how "clear visual clue" is defined or identified.

3. The benchmark does not effectively integrate multiple modalities. Although the authors claim that the three components are mutually supportive, the paper does not demonstrate how these components interact.

4. Experimental settings and evaluation metrics are crucial, yet placing them entirely in the appendix makes the paper hard to follow.

**Questions:**

The analysis section summarizes numerous issues. Among these, which problem is the most critical and has the greatest impact on the performance of current multimodal LLMs? Could resolving this issue potentially lead to the resolution of other problems?

---

> ### Author Response · Authors · 2025-11-26
>
> We thank the reviewer for the insightful comments. We address your concerns below.
>
> ---
>
> **Q1. Reorganizing the Introduction. Clarifying Motivation and Linking Research Gaps to Benchmark Design.**
>
> Thank you for your feedback, and we agree that writing can be improved. We have already made the change in the paper, and it is an easy fix for camera-ready.
>
> ---
>
> **Q2. Annotation Procedures and Inter-Annotator Agreement.**
>
> Our annotation process prioritized **accuracy verification** through a multi-stage consensus pipeline rather than a standard inter-annotator agreement (which is more suited to subjective tasks). 20 human annotators were involved.
> - **Initial Creation**: Annotators created tasks based on strict inclusion criteria mentioned in section 3.3.
>     - Each image contains at least one clear visual clue that the model can leverage through web search.
>     - Query must demand information outside the model’s prior knowledge or baseline visual reasoning; correctly answering requires some web search.
>     - A non-expert human with access to web search tools could accurately answer the question.
>     - The answer cannot be derived without consulting up-to-date external sources. They also should not collapse into pure visual reasoning tasks (e.g., OCR, object recognition, or simple counting).
>     - The image provides essential disambiguation. Without it, the query is unanswerable (e.g., asking “Who owns this team?” without showing a logo).
> - **Iterative Verification (Consensus)**: Rather than sampling a subset for agreement, we implemented a 100% cross-check. Every entry was reviewed by multiple distinct annotators. If an answer could not be consistently derived by the reviewer using search tools, the entry was flagged and either corrected or discarded.
> - **Completion**: This cycle continued until every problem in the final dataset had passed verification by every independent annotator with no outstanding flags. This ensures that all tasks in the benchmark have a verifiable, objective ground truth.
>
> ---
>
> **Q3. Defining the “Clear Visual Clue” in Vision Data.**
>
> We define a "clear visual clue" as an unambiguous visual anchor that links the image to the specific entity, location, or event required to resolve the query. During data curation, an image meets this criterion only if it contains features sufficient for a human expert to ground a search.
> These clues fall into two categories:
> - Explicit: Text, logos, specific uniforms, or product labels.
> - Implicit yet Distinctive: Unique architectural styles, layout arrangements, or specific color schemes combined with context.
>
> For example, Fig. 11 in the appendix exemplifies this definition. The image provides explicit evidence via background signage and implicit evidence via the players' jersey designs. While neither might fully identify the specific game in isolation, their combination forms a sufficient visual anchor to construct the search queries necessary to answer the prompt.

---

> ### Author Response · Authors · 2025-11-26
>
> **Q4. Integration and Interaction of the Three Benchmark Components Across Modalities.**
>
> We appreciate the reviewer’s suggestion regarding the integration of multiple modalities. While we agree that joint audio-vision processing is an important future direction, our decision to keep the suites distinct in MFCL was a deliberate design choice driven by **diagnostic precision** and **current model maturity**.
>
> **Avoiding Compounding Errors**: As detailed in Table 2, even SOTA models currently achieve only ~30% accuracy on vision-only tool use. Similarly, Table 1 demonstrates that audio noise alone causes significant degradation. Interleaving these modalities at this stage would likely result in compounding error rates that drive performance toward a non-informative floor. This would obscure the source of failure and prevent us from distinguishing between failures in acoustic perception, visual grounding, or tool-use reasoning.
>
> **(New Experiment) Validating Compounding Fragility**: To empirically validate this hypothesis, we ran an additional experiment using the Vision Base suite, but replacing text queries with Audio queries. Result attached below.
>
> | Model | Vision Base (Text Query) | Vision Base (Audio Query) |
> | :--- | :---: | :---: |
> | Gemini-2.5-Pro | 29.9 | 25.1 |
> | Llama-4-Maverick-17B-128E | 12.8 | 7.1 |
> | Qwen3-Omni-Flash | 16.5 | 9.6 |
>
> As hypothesized, the addition of the audio modality caused a distinct performance drop due to compounding perception errors. These results confirm that mixing modalities currently obscures the evaluation signal rather than clarifying it. By maintaining aligned but distinct suites, MFCL provides a gradient of difficulty that allows researchers to measure progress in visual grounding and acoustic robustness separately.
>
> **Community Accessibility**: Finally, we aim for MFCL to benchmark a broad spectrum of models. Currently, most flagship models (including GPT 5/4o, Claude Opus/Sonnet 4.5, and Grok 4, etc) do not yet support simultaneous Vision + Audio + Function Calling inputs. Requiring all three would restrict the benchmark to a small subset of models (e.g., Gemini 2.5 Pro, Llama 4), limiting its utility for the wider community. We view joint audio-vision items as a natural extension for the next version of MFCL once the field saturates the single-modality suites.
>
> ---
>
> **Q5. Relocation of Experimental Settings and Metrics to the Main Text.**
>
> Thank you for your feedback. We have made the change in the paper.
>
> ---
>
> **Q6. Identification of Critical Failure Modes and Their Downstream Impact.**
>
> Based on our error taxonomy (Section 5.1-5.2), across modalities, the most critical problem we observe is **unstable grounding of user intent** at the perception-to-tool-call boundary under noisy or ambiguous inputs. This single issue manifests differently in audio and vision, but in both cases it is what ultimately drives the largest performance drops; other failures are often downstream.
>
> **Audio suites**:
>
> **Competing Speech** is the dominant source of failure. Unlike stationary background noise, overlapping speakers sharply reduce *Intent Preservation* (FM1) by 20%. This is not just an ASR error: the model semantically blends background chatter with the user’s request, and then issues a syntactically valid but semantically wrong function call. Once the intent representation is corrupted, we frequently observe follow-on issues such as *Clarification Misfires* (FM4) and *Named-Entity Errors* (FM6).
>
> Thus, the “cocktail party” setting (separating user intent from competing talk) is the highest-leverage failure mode in audio. Improving robustness specifically to competing speech (e.g., explicit speaker/intent separation or better calibration around double-talk) would directly reduce the rate at which corrupted intents propagate through the rest of the pipeline and thereby mitigate several other failure modes.
>
> **Vision suite**:
>
> The analogous bottleneck is **Visual Reasoning Errors** (FM1/4/6), which arise not from low-level perception (OCR or object detection) but from how the model allocates attention and commits to a visual hypothesis. Models often overweight the wrong visual cue (FM1), prematurely anchor on an initial guess and resist updating (FM4), or abandon stronger alternatives they themselves surfaced (FM6). Once the visual grounding is wrong or unstable, downstream behaviors like *Poor Keyword Selection* and *Avoiding Tool Use* follow naturally: the tool-call pipeline is simply starting from the wrong visual hypothesis and is reluctant to self-correct.
>
> We therefore view “stable, self-corrective visual grounding” as the highest-leverage target on the vision side. Enhancing the model’s ability to (i) focus on the truly diagnostic visual cues, (ii) revise its hypothesis as new evidence appears, and (iii) explore competing visual interpretations before committing would propagate improvements to keyword selection, tool usage, and ultimately end-to-end accuracy.

---

### Official Review · Reviewer_ESaR · 2025-10-31

**Soundness:** 2
**Presentation:** 2
**Contribution:** 2
**Rating:** 4
**Confidence:** 4

**Summary:**

This paper introduces MFCL, a new benchmark for function calling in multimodal scenarios. The paper examines several cutting-edged models and reveals common failure patterns of these models, providing insights into developing multimodal agents.

**Strengths:**

- The topic (function-calling and multimodality) is timely.
- Comprehensive dataset design
- Clear description for the dataset construction

**Weaknesses:**

- Limited insight beyond enumeration of “failure modes.” Most FM categories merely restate known LLM limitations (ASR errors, over-reliance on text, conversational drift).
- Missing implementation details, such as the decoding hyperparameters. This may reduce the reproducibility of the paper.
- Statistical shallowness. Reported numbers are raw accuracies with no confidence intervals or significance testing.

**Questions:**

- How did you verify that the TTS-generated and noise-augmented audio realistically represents spontaneous human speech or real-world acoustic conditions? Was any human evaluation conducted to confirm perceptual naturalness?
- What procedures ensured the correctness and consistency of the expert-verified tasks? How many annotators were involved? What inter-annotator agreement was achieved?
- Given that the Vision set contains only 250 examples, why do you consider its coverage sufficient for robust evaluation?

---

> ### Author Response · Authors · 2025-11-26
>
> We thank the reviewer for the insightful comments. We address your concerns below.
>
> ---
>
> **Q1. Differentiation from Known LLM Limitations**
>
> Thank you for this feedback. We respectfully disagree that our taxonomy merely restates generic limitations. Our contribution lies in systematically mapping generic model failures to specific downstream consequences in the Multi-Modal Function Calling context, a distinction that existing benchmarks do not capture.
>
> We move beyond high-level categories to define actionable diagnostics:
> - **Audio Suite (Beyond “ASR Errors”)**: We decompose generic transcription errors into distinct execution pathologies. For example, *Intent Blending* and *Parameter Distortion* describe specific cases where background speech or microphone artifacts result in a **schema-valid but semantically incorrect API call** (e.g., executing book_flight(destination="Boston") instead of “Austin”). This is distinct from a generic ASR failure where the model simply outputs garbage; here, the tool-use capability masks the error, making it harder to detect. We also introduce Clarification Misfires, analyzed via our specific pipeline to distinguish harmless conversational turns from genuine failures.
> - **Vision Suite (Beyond “Over-reliance on Text”)**: We quantify failures based on the **downstream tool call**, not just the final answer. We identify Myopia (ignoring referenced background objects), Subset Confusion, and First-hop Bias. These are defined by which keywords are selected and whether a search is issued, offering granular insight into the decision-making process that standard VQA accuracy metrics miss.
> - **Interface-Specific Failures**: We demonstrate that these modes are interface-dependent. By contrasting native function-calling models with prompt-only baselines, we show that prompt-only models suffer from hallucinated visual capabilities (e.g., querying "square footage" assuming the tool can "see" the image), whereas native models exhibit different failure patterns.
>
> ---
>
> **Q2. Dataset & Source Code Release**
>
> All datasets and the full evaluation codebase are available in an open-source repository. To preserve double-blind anonymity during review, we provide an anonymized copy here: https://anonymous.4open.science/r/MFCL
> Our evaluation framework currently supports a broad range of models (100+ proprietary models and 90+ open-source models), which we believe will be a valuable contribution to the community. We used temperature=0 for all generations to ensure deterministic and reproducible evaluation. For all other hyperparameters, we use their default API settings.
>
> ---
>
> **Q3. Statistical Significance and Confidence Intervals**
>
> We thank the reviewer for emphasizing the importance of statistical rigor. We have updated Tables 1 and 2 to report 95% confidence intervals based on bootstrap resampling (n=10,000).
>
> ---
>
> **Q4. Synthetic Audio vs. Human-Recorded Speech for Real-World Audio Robustness**
>
> Our goal with MFCL True Audio is to approximate the range of conditions faced by deployed voice assistants, not to perfectly reproduce any one conversational corpus. We address realism at the following levels:
>
> - **On the text side**, we did not simply feed clean text to a TTS engine. As detailed in Section 3.1, we utilized a "Manager LLM" to inject 23 distinct types of spontaneous speech phenomena, including filled pauses ("um," "uh"), self-corrections ("no, sorry, I meant..."), and false starts.
> - **On the speech side**, we synthesize audio using a heterogeneous pool of neural TTS voices spanning multiple prosodies, accents, and speaking rates, which avoids overfitting to a single timbre or prosody profile.
> - **For acoustic conditions**, we mix speech with environmental noise and reverberation derived from realistic corpora (e.g., MUSAN and CHiME-5, which consists of real domestic recordings), and simulate channel effects such as packet loss, clipping, and room impulse responses. The SNR ranges and distortion strengths were initially tuned via iterative author listening so that utterances remained clearly intelligible while exhibiting plausible “busy café”, “noisy call”, or “reverberant room” characteristics.

---

> ### Author Response · Authors · 2025-11-26
>
> **Q5. Human Evaluation to Confirm Perceptual Naturalness**
>
> To directly verify perceptual naturalness, we conducted a human evaluation on our synthetic audio. We sampled 200 utterances (100 clean, 100 noise-augmented) and recruited 20 raters (internal volunteers). Each rater evaluated a random subset on a 1–5 scale (where 1 is worst and 5 is best) for: **Speech naturalness** (*“Does the voice sound like a plausible human speaker?”*), and **Acoustic realism** (*“Does the background / channel sound like a plausible real-world condition?”*; asked only for noisy clips). The results are shown below.
>
> | Metric | Clean Audio (Mean ± SD) | Noisy Audio (Mean ± SD) |
> | :--- | :---: | :---: |
> | **Speech Naturalness** | 4.1 ± 0.7 | 3.9 ± 0.8 |
> | **Acoustic Realism** | N/A | 3.8 ± 0.7 |
>
> In 83% of noisy clips, raters gave ≥3 on both speech naturalness and acoustic realism.
> These results indicate that listeners generally perceive our synthetic audio as natural and realistic.
>
>
> ---
>
> **Q6. Annotation Procedures and Inter-Annotator Agreement**
>
> Our annotation process prioritized **accuracy verification** through a multi-stage consensus pipeline rather than a standard inter-annotator agreement (which is more suited to subjective tasks). 20 human annotators were involved.
> - **Initial Creation**: Annotators created tasks based on strict inclusion criteria mentioned in section 3.3.
>     - Each image contains at least one clear visual clue that the model can leverage through web search.
>     - Query must demand information outside the model’s prior knowledge or baseline visual reasoning; correctly answering requires some web search.
>     - A non-expert human with access to web search tools could accurately answer the question.
>     - The answer cannot be derived without consulting up-to-date external sources. They also should not collapse into pure visual reasoning tasks (e.g., OCR, object recognition, or simple counting).
>     - The image provides essential disambiguation. Without it, the query is unanswerable (e.g., asking “Who owns this team?” without showing a logo).
> - **Iterative Verification (Consensus)**: Rather than sampling a subset for agreement, we implemented a 100% cross-check. Every entry was reviewed by multiple distinct annotators. If an answer could not be consistently derived by the reviewer using search tools, the entry was flagged and either corrected or discarded.
> - **Completion**: This cycle continued until every problem in the final dataset had passed verification by every independent annotator with no outstanding flags. This ensures that all tasks in the benchmark have a verifiable, objective ground truth.
>
> ---
>
> **Q7. Sufficiency of Vision Set Coverage for Robust Evaluation**
>
> We narrowed down to 250 examples as an initial set based on evaluating different models, capturing diversity, and considering the efforts in curating this dataset. The benefit of our open-source eval is that it is easily extensible by the community and experts.
>
> Appendix Section E (Figure 17) shows the topic breakdown of the vision image–query pairs, spanning over 12 domains.
>
> We also source images from multiple platforms (Google, Unsplash, Pexels, Reddit, and personal phone photos) to capture varied visual conditions and use cases:
>
> - Reddit provides diverse, realistic photos directly from personal devices, with natural variation in lighting, clutter, and everyday contexts.
>
> - Pexels and Unsplash contribute high-resolution, community-contributed images that are more polished and stylized, balancing the dataset with cleaner compositions.
>
> - Google is used to fill category gaps: as a search engine over the entire web, it is particularly useful for locating images for specific or design-focused categories (e.g., concerts, storefront layouts, canvas designs, and other niche scenes) that are underrepresented in curated platforms.
>
> - Personal photos further ground tasks in naturalistic environments.
>
> This combination is designed to cover a broad range of tool-using behaviors (from clean, “textbook” images to noisy, real-world scenes) despite the relatively small sample size.
>
> All our datasets are publicly available here: https://anonymous.4open.science/r/MFCL. We invite the reviewer to inspect the vision subset to verify the diversity and coverage we aim to achieve.

---

### Official Review · Reviewer_FoDx · 2025-11-01

**Soundness:** 3
**Presentation:** 4
**Contribution:** 3
**Rating:** 6
**Confidence:** 4

**Summary:**

The paper introduces MFCL, the first large-scale benchmark for evaluating multi-modal function calling (tool use) in large language models (LLMs). MFCL comprises 8.2K expert-verified tasks across three suites: True Audio, Text Audio, and Vision. Each example pairs a multi-modal user query (text, speech, or image) with a ground-truth toolcall trace, and includes controlled perturbations (accents, noise, occlusions, etc.) to stress the perception-to-action pipeline.

MFCL provides an automatic grader for exact-match scores on function names and arguments, enabling robust, reproducible evaluation without reliance on LLM judges. The authors benchmark leading models (e.g., GPT-4o, Gemini, Claude, GLM, xLAM, etc.), analyze failure modes (named-entity ASR errors, conversational drift, tool avoidance), and present a taxonomy to guide future research. The dataset, taxonomy, and diagnostics are released to accelerate progress on reliable multi-modal agents.

**Strengths:**

The strengths of the paper are:

1. Originality:
    - First benchmark to systematically evaluate multi-modal function calling under real-world perturbations.
    - Introduces controlled perturbations and a taxonomy of failure modes.
    - Unifies text, audio, and vision evaluation in a single framework.

2. Quality:
    - Expert-verified tasks, realistic data augmentation, and comprehensive error analysis.
    - Automatic grading at function and argument level, enabling reproducible and robust evaluation.
    - Strong experimental design, with ablations and comparisons across many models.

3. Clarity:
    - Clear motivation, methodology, and results presentation.
    - Figures and tables directly support claims; taxonomy is actionable.

4. Significance:
    - MFCL will become a standard for evaluating multi-modal tool-augmented agents.
    - The insights into failure modes and robustness are valuable for both research and deployment.

**Weaknesses:**

No major weaknesses. The study of a multi-modal functional calling benchmark is very useful for developing agentic LLM in real-world scenarios.

**Questions:**

Minor questions:

1. The failure mode analysis is very interesting. Did authors have quantitative results in addition to the qualitative examples?
2. Any plan for the release of the benchmark?
3. Have you/ do you have plans to evaluate smaller models on the benchmark? like Qwen-omni, and other multi-modal LLMs with similar size?

---

> ### Author Response · Authors · 2025-11-26
>
> We thank the reviewer for the insightful comments. We address your concerns below.
>
> ---
>
> **Q1. Quantitative Results for Failure Mode Analysis**
>
> We report the distribution of the different failure modes in Figure 5. For readability, we merged some closely related failure modes into aggregated labels in the chart; we have now added an explicit explanation of this mapping in the main text. For example, in the vision plot, failure modes 4 and 6 are subsumed under “visual reasoning” because both arise from early perceptual missteps that either lock attention onto an irrelevant region or prematurely discard informative cues, ultimately derailing the reasoning trajectory.
>
> ---
>
> **Q2. Dataset & Source Code Release**
>
> Yes. All datasets and the full evaluation codebase are available in an open-source repository. To preserve double-blind anonymity during review, we provide an anonymized copy here: https://anonymous.4open.science/r/MFCL
> Our evaluation framework currently supports a broad range of models (100+ proprietary models and 90+ open-source models), which we believe will be a valuable contribution to the community.
>
> ---
>
> **Q3. Plans for Evaluating Smaller-Scale Multimodal LLMs (e.g., Qwen3-Omni)**
>
> Yes. Qwen3-Omni series was released after our ICLR submission deadline, so we could not include them in the submission. We have added them and included the evals in the paper.

---

### Note · Program_Chairs · 2026-01-17
**Submission Desk Rejected by Program Chairs**

The following references in this submission do not refer to real documents and/or have major errors in bibliographic information:

 Haotian Liu et al. Structured text generation with jsonformer. arXiv preprint arXiv:2310.10648, 2023.

Guanghui Qin et al. Toolbench: A comprehensive evaluation of tool-augmented large language models. In Advances in Neural Information Processing Systems, 2023.

Viraj Gudibande et al. False-positive and hallucination analysis in large language models. arXiv preprint arXiv:2311.00582, 2023.

Haotian Liu et al. Mmbench: Evaluating multimodal models on comprehensive tasks. In ECCV, 2024.

Kai Sun et al. LlamaGuard: Safety alignment for large language models. arXiv preprint arXiv:2312.04209, 2023.